# Photo-Curing Chitosan-g-N-Methylolacrylamide Compositions: Synthesis and Characterization

**Sergey Uspenskii** [1,*], **Vladislav Potseleev** [2], **Eugenia Svidchenko** [1], **Galina Goncharuk** [1], **Alexander Zelenetskii** [1] **and Tatiana Akopova** [1,*]

1   Enikolopov Institute of Synthetic Polymeric Materials, Russian Academy of Sciences, 70, Profsoyuznaya Str., 117393 Moscow, Russia
2   Faculty of Chemistry, Moscow State University, 1, Leninskie Gory, 119991 Moscow, Russia
*   Correspondence: uspenskii@ispm.ru (S.U.); akopova@ispm.ru (T.A.); Tel.: +7-(916)701-13-12 (S.U.); +7-(903)223-76-12 (T.A.)

**Abstract:** Chitosan is one of the promising compounds for use in various fields of medicine. However, for successful application, materials based on it must be insoluble in water and have specified physical and mechanical properties. In this work, we studied the interaction of *N*-methylolacrylamide (NMA) and chitosan upon concentration of the solutions, both under the action of UV radiation and without it, which results in curing of the polymer matrix. The main products, proposed mechanisms of the crosslinking reaction, and the influence of external conditions on these processes have been revealed using NMR, IR, and UV spectroscopy. It was found that the reaction proceeds along three pathways. The main reactions proceed with the amino groups of chitosan, and the hydroxymethyl and vinyl groups of NMA. Studies have shown that for the formation of insoluble materials based on chitosan, the best content in the initial cast solution is 2 wt% of chitosan at 0.25 wt% concentration of NMA. Films formed from such solutions possessed high strength and deformation characteristics, namely an elastic modulus of about 1500 GPa, a strength of about 30 MPa, and an elongation at break of about 100%.

**Keywords:** chitosan; *N*-methylolacrylamide; films; UV irradiation; reaction mechanism; strength and deformation characteristics; swelling; viscosity

## 1. Introduction

Chitosan is a well-known polysaccharide that has antimicrobial activity and is biocompatibile with most tissue types. Chitosan-based materials have a high potential for their application in pharmaceuticals, medicine, and tissue engineering [1–3]. The processing of chitosan into products is carried out by dissolving it, and dilute aqueous solutions of acids are usually used as a solvent. Therefore, products formed from chitosan solutions are water-soluble salts of chitosan. However, the material must be water insoluble (with limited swelling) but degradable in a number of applications, namely tissue engineering constructions and surgical threads, in the case of textile threads, etc. To obtain water-insoluble chitosan products and control their plasticity, the polymer matrices should be modified. Usually, epichlorohydrin, glutaraldehyde, genipin, and ethylene glycol diglycidyl ether are added to the chitosan molding solution to achieve curing [4–6]. As a result, a chemically crosslinked gel is formed, which is suitable for only a limited range of products. For example, it becomes difficult to carry out the process of molding film and fibrous materials without destroying the gel.

To prevent uncontrolled chemical crosslinking, it is necessary that the chitosan crosslinking process be completed after fiber coagulation in a precipitation bath during "wet", "dry", and "dry-wet" casting a polymer solution [7–9] or electrospinning [10]. This approach can be achieved by using chitosan-based photopolymerizing compositions as the initial

molding solution. The first step towards the implementation of this approach is the preparation of a modified chitosan containing substituent in the main or side polymer chain that can be involved in polymerization under UV exposure. The choice of components for the synthesis of derivatives and copolymers is limited by the requirement of their biocompatibility and, in the case of their application as a material for regenerative medicine, by their ability to biodegrade. The research literature focuses on chitosan and 1,4-butanediol diglycidyl ether or glycidyl methacrylate [11–15] and, to a lesser extent, on chitosan and NMA-based compositions [16,17]. However, a detailed description of possible ways of interaction upon concentration of acidic aqueous solutions of chitosan containing NMA has not been considered previously. The purpose of this work is the development and scientific justification of a method for obtaining biocompatible chitosan hydrogels using NMA and modeling the curing of a polymer matrix under the action of ultraviolet radiation.

## 2. Materials and Methods

### 2.1. Materials

Chitosan with a molecular weight of 100 kD and a degree of deacetylation of 86% was purchased from Acros Organics. *N*-methylolacrylamide was purchased from Sigma-Aldrich (*N*-hydroxymethyl-acrylamide solution, about 48% in water). Glacial acetic acid was purchased from ChemMed (Moscow, Russia).

### 2.2. Preparation of Chitosan Solutions

The solutions were prepared by preliminary swelling of chitosan in water and subsequent dissolution by adding glacial acetic acid (AcOH) to achieve a concentration of the polymer of 2 wt%.

The solutions were divided into five parts for further mixing with the calculated amounts of NMA, as shown in Table 1, and kept at room temperature for 24 h for deaeration.

**Table 1.** Composition of the cast solution based on chitosan and NMA.

| Components | Experiment Number | Experiment 1 | Experiment 2 | Experiment 3 | Experiment 4 | Experiment 5 |
|---|---|---|---|---|---|---|
| | | Chitosan 2 wt% solution, g | Chitosan and NMA solution 1, g | Chitosan and NMA solution 2, g | Chitosan and NMA solution 3, g | Chitosan and NMA solution 4, g |
| Chitosan | | 0.52 | 0.104 | 0.104 | 0.104 | 0.104 |
| NMA 48% | | - | 0.0052 | 0.0217 | 0.043 | 0.065 |
| AcOH 100% | | 0.72 | 0.144 | 0.144 | 0.144 | 0.144 |
| $H_2O$ | | 24.48 | 4.896 | 4.896 | 4.896 | 4.896 |

### 2.3. Viscosity Measurements

The viscosity of chitosan solutions was measured on an AND SV-10 vibroviscometer (Japan) at 25 °C.

### 2.4. Film Casting

The films were formed by solvent evaporation at room temperature. The thickness of the films depends on the mass of the solution evenly distributed on the polystyrene Petri dish. The Formula (1) for calculating the mass of the solution (*m*, g) that is needed to obtain films of the required thickness was the following:

$$m = \frac{S \times h \times \rho}{C},$$

(1)

where *S* is the substrate area, $cm^2$; *h* is the film thickness, cm; $\rho$ is the polymer density, $g/cm^3$, (for chitosan, 1.44); and *C* is the concentration of the solution, wt%.

The films were made from chitosan solutions with different contents of NMA and from the same solution but irradiated with UV.

The quality control of washing with salt (sodium acetate) was carried out using spectrophotometry by analyzing wash water.

### 2.5. Conditions of Ultraviolet Irradiation of Chitosan Solutions with Different NMA Content

The prepared solutions were exposed to ultraviolet radiation (UV). The irradiation duration was 30, 60, 120, and 240 min with an unfiltered parallel beam of light from a DRSh-500 mercury lamp. The photon flux was $1 \times 10^{17}$ photon/(cm$^2$×s). The scheme of the irradiation installation is shown in Figure 1.

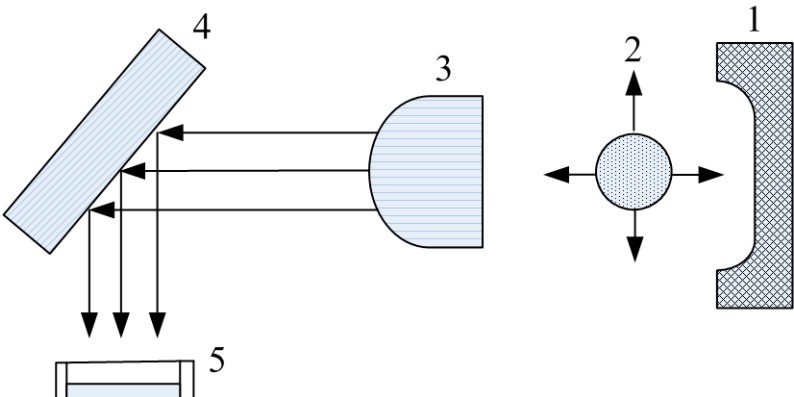

**Figure 1.** The scheme of the irradiation installation; 1—reflector, 2—mercury lamp, 3—focusing lens, 4—rotary mirror, and 5—Petri dish with sample.

The light from the lamp (2) was collected, focused, and directed to the sample in a Petri dish (5) using a reflector (1), a focusing lens (3), and a rotary mirror (4). All samples were irradiated (Experiment 2, 3, 4, 5; Table 1). The films made of them were compared with non-treated ones using relevant technical/spectroscopic data.

### 2.6. FTIR-Spectroscopy

FTIR spectra were recorded on a Bruker Vertex 70 spectrometer (USA). All spectra were initially collected in ATR mode at a resolution of 4 cm$^{-1}$ by employing an ATR-mono-reflection Gladi ATR (Pike Technologies, Madison, WI, USA) accessory equipped with a diamond crystal (n = 2.4; angle of incidence 45 deg.). The obtained ATR spectra were converted into IR-Absorbance mode. All the spectra presented in this work were recorded and treated using a set of programs: Bruker Opus (version 6.1).

### 2.7. NMR Spectroscopy

The $^1$H NMR spectra were recorded on a Bruker 300 MHz UltraShield spectrometer at 20 °C. The spectra were processed using the ACD/Labs program. Samples were prepared as follows: 30 mg of polymers were dissolved in 500 μL of a 2 wt% HCl solution. The resulting solutions were poured into individual ampoules using an elongated capillary.

### 2.8. UV/Vis-Spectrophotometry of Chitosan-NMA Films

The absorption spectra of the pure chitosan and NMA crosslinked chitosan (chitosan-NMA) films were recorded on a standard Shimadzu UV-2501PC dual-beam spectropho-tometer (Kyoto, Japan) in the spectral region from 190 to 800 nm. Measurements were performed relative to air.

### 2.9. Optical Microphotographs of Chitosan-NMA Films

The optical microphotography of the films was obtained using a Levenhuk digital monocular microscope (USA). Magnification of ×400 was used.

*2.10. The Study on the Swelling Behavior of the Films*

The absorption of moisture by the films was studied by the gravimetric method. The film samples were kept in water for various times; the films were dried with filter paper before weighing. The degree of swelling ($\alpha$, %) was calculated as following:

$$\alpha = \frac{m - m_0}{m_0} \times 100\%, \tag{2}$$

where $m_0$ is the weight of the original film, g and $m$ is the weight of the film after swelling, g.

According to the data obtained, the dependence of the degree of swelling on time $\alpha = f(\tau)$ was plotted, and the maximum value $\alpha$ was determined. The measurement results are given only for the maximum value of $\alpha$. The standard deviation for all points on the graphs was 0.01 for 5 parallel measurements.

*2.11. Determination of Stress-Strain and Strength Characteristics of Films*

The strain-strength curves of the initial and modified chitosan films with known thickness were obtained using a Shimadzu Autograph AGS 20 kN10 tensile machine (Japan). The starting length of the samples was 70 mm, and the tension rate was 20 mm/min. The relative tensile load and relative tensile elongation were measured. All samples of the films were preconditioned at 45% moisture content (films were kept in a desiccator over saturated aqueous $NaNO_2$ for one week) and 100% (see Section 2.10). Mechanical characteristics of the films, namely tensile strength, elastic modulus, and elongation at break, were calculated as the average value of four measurements of a film sample, taking into account its thickness, using the instrument software.

## 3. Results and Discussion

It is known that with an increase in the concentrations of reagents, the rate of their interaction increases, for bimolecular reactions, exponentially. However, in photo-initiated reactions, the interaction scheme is complex, and the polymer concentration must be low for radiation to penetrate. The transition from the regime of diluted solutions to the moderately concentrated ones can be established more reliably using the modified Vinogradov–Pokrovskii method [18]. However, to achieve the goals of this work, it is sufficient to establish an approximate range of concentrations for the formation of a fluctuation network. The polymer concentration needed to start the crosslinking when the polymer macromolecules begin to associate was determined from the viscosity concentration curves (Figure S1). It follows from the experiments that the concentration at which the network is formed corresponds to $2.0 \pm 0.2$ wt%. Therefore, all further experiments were done with 2.0 wt% solutions of chitosan.

*N*-methylolacrylamide was chosen as a model object for studying chitosan modification by photosensitive unsaturated acrylates (Figure 2).

**Figure 2.** Structural formula of *N*-methylolacrylamide.

The presence in the structure of two functional groups (*N*-methylol and terminal vinyl) makes it an interesting object for studying the interaction of chitosan with bifunctional monomers.

At the second stage of the work, it was necessary to determine the conditions under which NMA reacts with chitosan. For this, chitosan was mixed with NMA in various ratios (experiments 2—5; Table 1) in an acidic medium with stirring. After an hour of mixing

chitosan with a given amount of NMA, the samples were analyzed using NMR spectral analysis (Figure S2). The spectrum shows absorption bands characteristic of chitosan and NMA—$\delta$ = 4.5 and 4.8 ppm (1H, H—C$^1$), related to acylated and deacylated units, respectively; 3.1 (1H, H—C$^2$); 3.5–4.0 (5H, H—C$^{3,4,5,6}$) ppm. The bands corresponding to the products were not detected, indicating the absence of interaction.

At the third stage, films were formed from similar solutions (experiments 2—5; Table 1), which were kept for 48 h. The resulting films in the salt form of chitosan were washed with chloroform to remove NMA oligomers. After drying, the films lost their solubility in dilute acid solutions, which confirmed the crosslinking reaction. The yield of chitosan ranged from 92 to 94%.

Figure 3 presents the IR spectra of the obtained films. The spectra are normalized to the 1153 cm$^{-1}$ ($\nu$CO) band of the asymmetric glycosidic bond. The numbers of spectral bands in Figure 3 correspond to the numbers of experiments in Table 1.

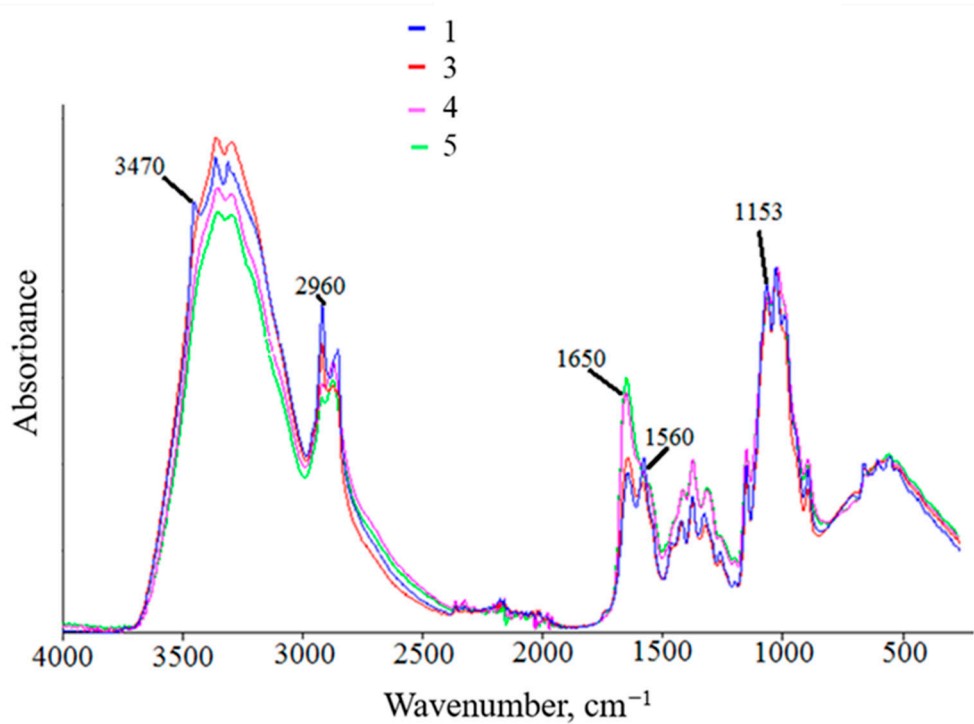

**Figure 3.** IR spectra of the samples with various contents of NMA: 0 wt% (1, initial chitosan sample); 10 wt% (3); 20 wt% (4); 30 wt% (5), the sample number corresponds to those of experiments in Table 1.

In the region of 1650 cm$^{-1}$, carbonyl stretching vibrations characteristic of chitosan (C=O) are observed. This band indicates the presence of chitin units in the initial polymer. With increasing NMA concentration, the intensity of the band increases, indicating the increase in concentration of the amide group. Since the product loses solubility, it can be assumed that both groups of NMA are involved in the reaction. Otherwise, the products of chitosan and NMA interaction should be soluble in acid solutions.

Proposed scheme of chitosan interaction with hydroxymethyl groups and vinyl groups of NMA is presented in Figure 4. These reactions are confirmed by the IR spectra, in which the characteristic bands for the primary amino group, namely stretching asymmetric vibrations ($\nu$NH2, 3470 cm$^{-1}$) and bending vibrations ($\delta$NH2, 1560 cm$^{-1}$), completely disappear in the spectra of the products when the reagent content is high enough. The latter appear in the spectrum at 1600 cm$^{-1}$ only as a shoulder to the intense absorption band of carbonyl stretching vibrations. However, the overall decrease in the intensity of the band in the region of 3400–3000 cm$^{-1}$ indicates that the reaction (2) with the hydroxyl groups of chitosan occurs, resulting in the formation of esters.

**Figure 4.** Proposed scheme of chitosan interaction with NMA hydroxymethyl groups (1, 2); NMA vinyl groups (3).

Stretching vibrations ($\delta$CH3) in the region of 2960 cm$^{-1}$ are observed in the spectra. This band is attributed to chitin units of chitosan. With increasing NMA concentration, the intensity of this band decreases. This can occur as a result of acid-catalyzed deacetylation of chitin units in the chitosan structure. The shape of the doublet changes due to the presence of additional methylene groups coming from the NMA in the product structure.

Presumably, reactions (1) and (2) proceed according to the $S_N1$ and $S_N2$ mechanisms. Figure S3 shows a diagram of the possible reaction mechanism. The hydroxyl group, due to the lone electron pair of the oxygen atom, acts as a weak base (proton acceptor), so the protonation of the hydroxyl group occurs in the presence of acids (the formation of an O-H donor-acceptor bond). Then, at the first stage, the alkyl hydroxonium cation splits off water, turning into a carbocation. This stage is rate-limiting. At stage 2, carbocation "quickly" causes a heterolytic bond cleavage in the nucleophile with the release of a proton, resulting in the reaction product. Figure S4 shows the possible mechanism of reaction (1), i.e., $S_N2$ bimolecular nucleophilic substitution. This mechanism consists of the almost simultaneous elimination of water and the addition of a nucleophile. The reaction proceeds in one step, through a transition state. The rate of the reaction depends on the concentration of the two reagents, so the mechanism is called bimolecular substitution. Preferably, the reaction with an amino group will occur due to its greater nucleophilicity. Since the above reactions proceed in an aqueous medium, which contributes to the stabilization of the carbocation and reduces the activity of the nucleophilic reagent due to the solvation of a lone electron pair, the $S_N1$ mechanism seems more likely [19]. The NMA molecule has a reactive acrylic group. Figure S5 presents the proposed reaction (3) mechanism. The first step is the protonation of the oxygen atom. Then a carbocation is formed. This carbocation is more stable due to the fact that the charge is distributed along the length of the chain from the carbonyl and the electron deficit of the carbon atom is greatest. The keto-enol equilibrium is also present in this system, and the nucleophilic amino group of chitosan interacts with the ketone form with the release of the initial proton (catalyst). The driving force of this reaction is the formation of a new carbon-carbon bond with an enthalpy factor of at least 330 kJ/mol. Such an enthalpy factor is sufficient to compensate for the decrease in entropy during the formation of one particle from two.

Figure 5 shows the proposed structure of the crosslinked chitosan-g-*N*-methylolacrylamide system. To obtain this structure, both NMA groups need to react with different chitosan molecules.

**Figure 5.** Proposed structure of the crosslinked copolymer of chitosan and NMA.

The effect of UV radiation on the curing (polymerization) of the composition was studied by examining the results of irradiating this composition with an unfiltered parallel beam of light from a DRSh-500 mercury lamp. Films were formed from the irradiated solutions, which were kept for 48 h. Then the films were washed with water to remove salts and with chloroform to remove NMA oligomers. The films partially lost their solubility (6–8% mass loss was fixed) in dilute acid solutions, which confirms the crosslinking reaction. The sol fraction collected after washing the products with chloroform (3–5 wt%) was analyzed by NMR spectroscopy (Figure 6). These are non-crosslinked products, but they also indicate the presence of an interaction since they contain chitosan, which is insoluble in chloroform if unmodified.

Spectrum (A) corresponds to the spectrum of pure chitosan and all peaks are assigned according to literature sources. The position and intensity of the signals in the $^1$H NMR spectra correspond to the structure of chitosan and are consistent with the literature data [20–22]: $\delta$ = 4.5 and 4.8 ppm (1H, H—C$^1$), related to acylated and deacylated units, respectively; 2.7 (1H, H—C$^2$); 3.2–3.7 (5H, H—C$^{3,4,5,6}$); 1.7 (3H, NHCOOCH$_3$) ppm. Spectrum (B) corresponds to the sol fraction of crosslinked chitosan films. Signals at 5.4 and 5.9 ppm are related to the terminal and internal atoms of the NMA unsaturated group. There are also several signals in the region of 8.2–6.7 ppm corresponding to the amide groups of chitosan (8.2 ppm) and the products of its interaction with NMA, suggestive of polyacrylate fragments in their structures when UV irradiation was applied.

Thus, it can be assumed that the structures (A and B) illustrated in Figure 7 are present in the insoluble part (films) and that in the system A, NMA interacts with chitosan via the S$_N$1 and S$_N$2 mechanisms (the Michael reaction), while in the system B, S$_N$1 and S$_N$2 substitutions with subsequent polymerization (crosslinking) of grafted copolymers prevail.

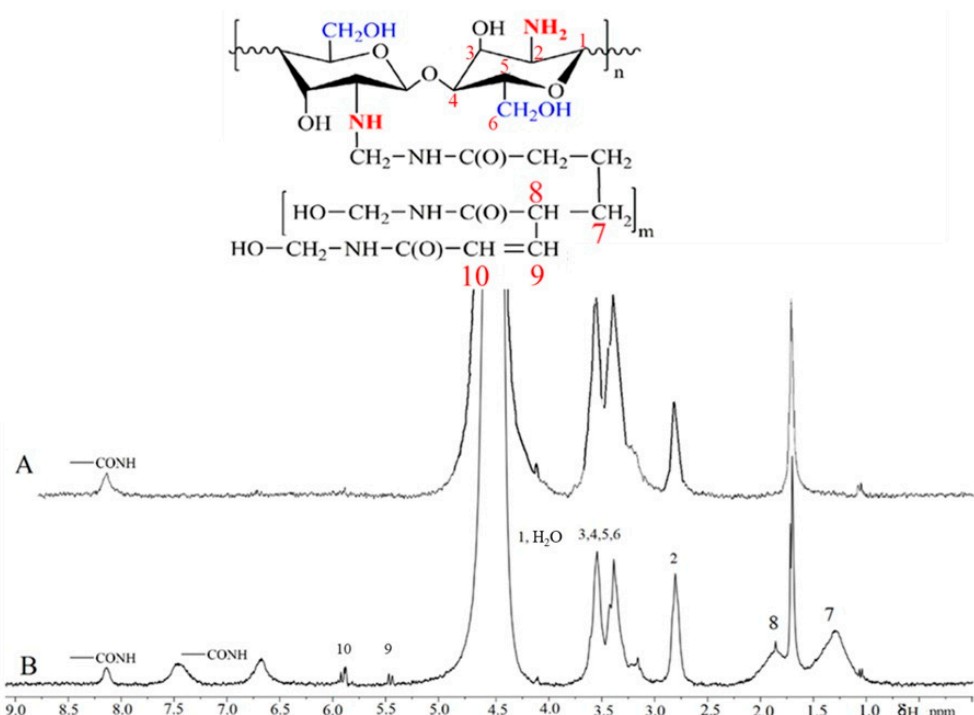

**Figure 6.** $^1$H NMR spectra of pure chitosan (**A**); soluble fraction of the irradiated samples (**B**).

**Figure 7.** Proposed crosslinked structures of chitosan-g-*N*-methylolacrylamide obtained under different conditions ((**A**)—concentration, (**B**)—UV irradiation).

Changes in the structure of chitosan affect the optical properties of the films (Figure 8). The data obtained show that an increase in the NMA content is accompanied by a synchronous shift of the short-wavelength edge of the absorption band to the long-wavelength region (from 250 to 320 nm). The film with the highest NMA content was cloudy, inhomogeneous, and highly scattered. The micrographs (Figure 9) demonstrate that after drying, NMA isolates into a separate phase.

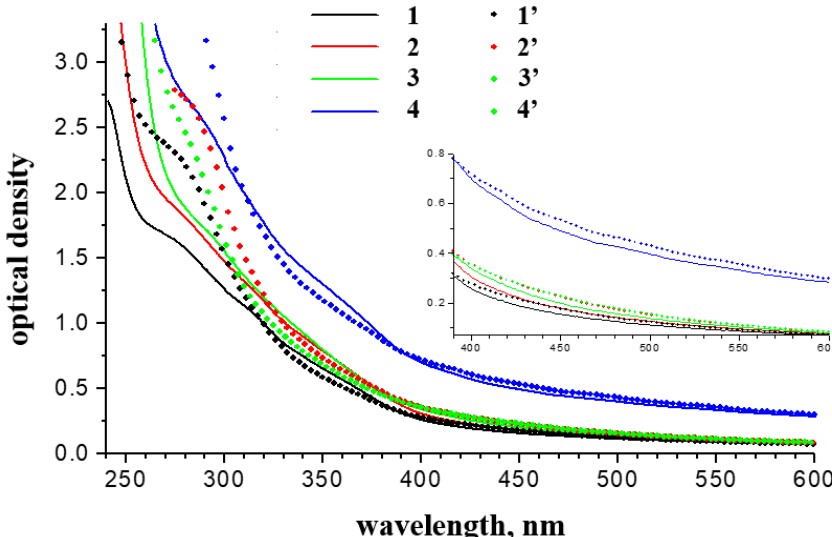

**Figure 8.** Absorption spectra of films from chitosan with content of NMA: 5 wt% (1, 1′); 10 wt% (2, 2′); 20 wt% (3, 3′); 30 wt% (4, 4′). 1–4—before (without) UV exposure; 1′-4′—after UV exposure (1 h). The inset shows the spectral region 390–600 nm. The film thickness was about 70 μm.

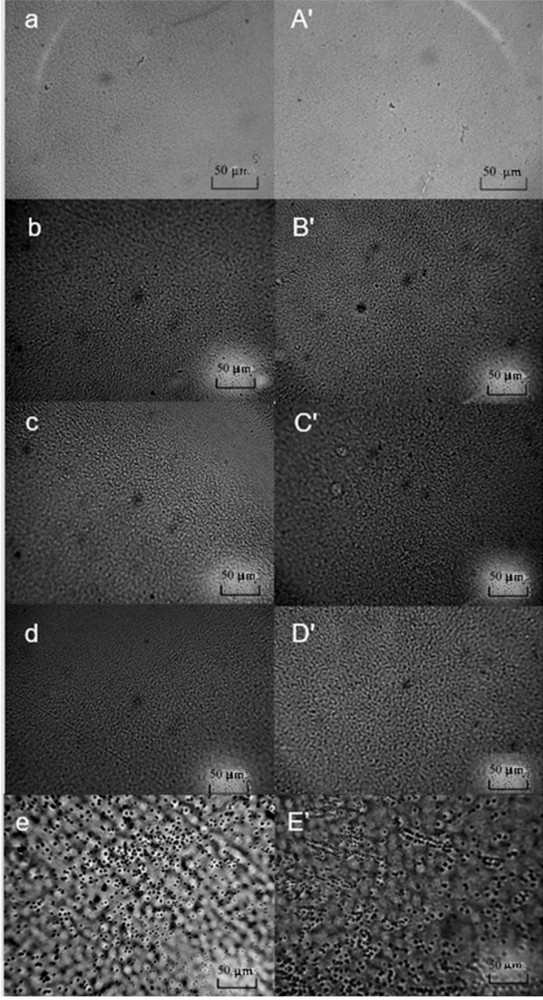

**Figure 9.** Optical micrographs of films from chitosan with content of NMA: 0 wt% (**a**, **A′**); 5 wt% (**b**, **B′**); 10 wt% (**c**, **C′**); 20 wt% (**d**, **D′**); 30 wt% (**e**, **E′**). a, b, c, d, e—before (without) UV exposure; A′, B′, C′, D′—after UV exposure.

Irradiation leads to similar changes in the absorption spectrum for all films: there is a decrease in the intensity of the bands related to the original chitosan (in the region of 320–390 nm) and an increase in the intensity of the bands in the region of 250–320 nm. A new absorption band appears in the region of 390–550 nm, which is characteristic of crosslinks in chitosan and its copolymers (Figure 8, inset). These changes may be associated with the appearance of new chromophores as a result of the formation of crosslinks between the original chromophore groups.

The micrographs (Figure 9) show the presence of the inclusions in all films. The film with the highest NMA content also shows the isolation of the NMA into a separate phase with inclusion sizes of 3–5 µm. At higher magnification, in all films except the last one, sections with different refractive indices are visible, distributed over the entire field of view. The size of these areas (which are visually darker) is about 2 µm.

Depending on the conditions of crosslinking and the ratio of chitosan: crosslinking reagent, the degree of crosslinking of the chitosan macromolecules can be varied. A change in the degree of crosslinking leads to a change in the degree of swelling (moisture absorption, index $\alpha$) of the films (Figure 10). With an increase in the content of the NMA, the sorption capacity and the degree of swelling of chitosan increase and reach equilibrium for non-irradiated chitosan films containing 20 wt% NMA. However, for the similar chitosan: NMA ratios, the sorption capacity of the modified chitosan films decreases with an increase in the content of NMA in the films (Figure 10 (1)). This indicates an increase in the network density with the introduction of NMA into chitosan and UV (Figure 10 (2)) irradiation. The optimal sorption capacity has a sample containing 12.5 ± 0.5 wt% NMA (Figure 10, circled area). It can be assumed that this sample will be characterized by higher deformation and strength characteristics.

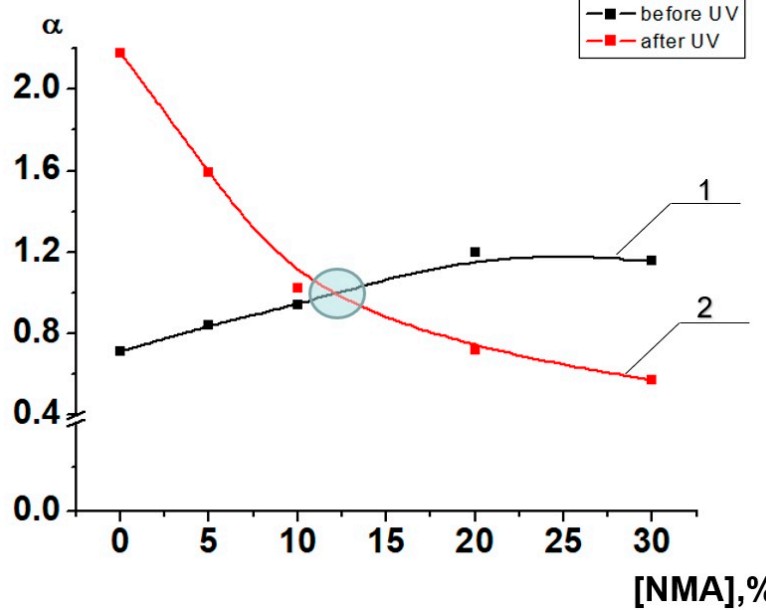

**Figure 10.** Dependence of the maximum degree of swelling in water ($\alpha$) of chitosan films crosslinked with NMA in the NMA concentration range of 0–30 wt%: (1) without UV irradiation; (2) in the presence of UV irradiation. The standard deviation for all points on the graphs was 0.01 for 5 parallel measurements.

The deformation-strength characteristics of chitosan films containing various amounts of NMA are presented in Table 2. An increase in the degree of modification of chitosan by NMA fragments leads to a decrease in their strength and elasticity. Films become significantly brittle after UV irradiation. At the same time, chitosan films modified with

NMA have moisture absorption, which, with maximum water sorption, contributes to an increase in elasticity by several times.

**Table 2.** Strain and strength characteristics of the films from chitosan with a content of NMA of 5–30 wt% obtained at different conditions.

| [NMA], % | Strength, MPa | Elongation, % | Module, GPa |
|---|---|---|---|
| Before (without) UV irradiation, 45% moisture | | | |
| 0 | 40.8 | 37.3 | 2067 |
| 5 | 39.3 | 29.3 | 2160 |
| 10 | 36.9 | 29.2 | 1774 |
| 20 | 27.7 | 28.7 | 1526 |
| 30 | 36.2 | 34.6 | 1353 |
| After UV irradiation, 45% moisture | | | |
| 0 | 40.2 | 5.4 | 1847 |
| 5 | 39.0 | 6.6 | 1835 |
| 10 | 36.9 | 8.9 | 1849 |
| 20 | 27.7 | 10.6 | 1806 |
| 30 | 36.2 | 10.9 | 1726 |
| After UV irradiation, 100% moisture, maximum swelling | | | |
| 0 | - | - | - |
| 5 | 36.0 | 78.6 | 1532 |
| 10 | 30.1 | 96.2 | 1507 |
| 20 | 23.7 | 102.4 | 1498 |
| 30 | 20.9 | 105.7 | 1474 |

## 4. Conclusions

In this work, the crosslinked chitosan-g-*N*-methylolacrylamide copolymers were obtained by concentrating acidic aqueous solutions of chitosan containing NMA both under the action of UV radiation and without it. Possible ways of interaction between NMA and chitosan through the mechanisms $S_N1$, $S_N2$, and nucleophilic addition at the conjugated vinyl group are discussed. The proposed mechanisms are classical. It is difficult to prove them fully in the resulting cross-linked products (which, however, were carefully purified from unreacted non-cross-linked components), but we did not set such a goal. At the same time, understanding the ongoing processes allows us to control the properties of the materials. The chemical structure of the resulting products was confirmed using IR, UV, and NMR spectroscopy. It is assumed that UV exposure leads to the formation of cross-links at the NMA acrylate groups, while a simple concentrating of the solutions is dominated by the nucleophilic addition of acrylic groups to the primary amino groups of chitosan. The material properties were carefully characterized depending on the ratio of reagents (degree of swelling, morphology, and mechanical characteristics). The optimal ratio of chitosan: NMA in the composition of chitosan solutions was found to be 10:1. With this ratio, moisture absorption, strength, and elasticity (elongation at break) of the films were maximized. The developed methods can be applied in biomedicine, where UV initiation is used, in particular, in two-photon stereolithography to produce hydrogels (scaffolds) of complex architecture.

**Supplementary Materials:** The following supporting information can be downloaded at: https://www.mdpi.com/article/10.3390/polysaccharides3040049/s1, Figure S1: The dependence of the dynamic viscosity of chitosan solutions on the concentration, Figure S2: NMR spectrum of the freshly prepared solution of NMA and chitosan. Figure S3: $S_N1$ mechanism of interaction of chitosan and NMA. Figure S4: $S_N2$ mechanism of the interaction between chitosan and NMA. Figure S5: Proposed scheme of nucleophilic addition mechanism.

**Author Contributions:** Conceptualization, S.U. and T.A.; methodology, S.U. and E.S.; validation, E.S. and A.Z.; investigation, S.U., E.S., V.P. and G.G.; data curation, S.U.; writing—original draft preparation, S.U. and V.P.; writing—review and editing, S.U., T.A. and A.Z.; visualization, S.U.; supervision, T.A. and A.Z.; project administration, S.U.; funding acquisition, A.Z. All authors have read and agreed to the published version of the manuscript.

**Funding:** This work was supported by Grant-in-Aid for Scientific Research Ministry of Science and Higher Education of the Russian Federation, grant No. FFSM-2021-0006.

**Institutional Review Board Statement:** Not applicable.

**Informed Consent Statement:** Not applicable.

**Data Availability Statement:** The data presented in this study are available upon request from the corresponding author.

**Acknowledgments:** NMR spectra registration was performed with the financial support of the Ministry of Science and Higher Education of the Russian Federation using the equipment of the Collaborative Access Center "Center for Polymer Research" of ISPM RAS.

**Conflicts of Interest:** The authors declare no conflict of interest.

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
