# Peer review of "Photo-Curing Chitosan-g-N-Methylolacrylamide Compositions: Synthesis and Characterization"

_2673-4176, doi:10.3390/polysaccharides3040049_

Round 1

Reviewer 1 Report

The paper could be interesting, if the paper would be explain objectives, methodology and tittle of all figures better, it is very difficult to understand, the objective the experimental part and the conclusions, also the mechanisms; because they don’t explain if they are talking about UV or thermal reactions.

-Abstract. It said:  In this work the N-methylolacrylamide (NMA) and chitosan interaction under the action of UV radiation and heat treatment has been studied. But-In Introducction page 2 line 50-52 said: The purpose of this work is the development and scientific justification of a method for obtaining biocompatible chitosan hydrogels using NMA and modeling the curing of a polymer matrix under the action of ultraviolet radiation.  -It is not clear if they wanted compare heat with UV crosslinking reactions or they wanted to use both in the reaction.

-Page 2 line 54 in section 2.2  It is not clear through the text if it was UV irradiated solutions of CS and also CS films. It is necessary to explain it well.

-Page 2, line 72. In which substrate?

-Page 3 it said. The films were made of all irradiated samples and the same samples but not UV irradiated. please clarify.

-In other part of the text said modification of CS in solution is most effective carried out when polymer macromolecule forms a network. What kind of modification? Modification was carried out in solution and in film?, Explain better it through the text.

-page 3 line 92. All spectra were initially collected in ATR mode. Initially? and then in what mode?. And better to change the word collected by characterized or determine.

Page 4 line 130 said: The polymer concentration at which the macromolecules began to form an engagement network….Better to said: the polymer concentration needed to start the crosslinking.

-Page 4 line 131. The author said: increase the viscosity due to structuring of solutions’?’ it is necessary to clarify it.

-Page 5, line 148. NMA reacts with chitosan? How reacts? By crosslinking by copolymerization or what?

-Fig 1. Author said. Dependence of viscosity on concentration from section II, characterized by smooth increase in viscosity due to structuring region??? The paragraph is not clear enough, it is necessary to explain better.  What kind of structuring?. It is necessary to do the figure better.

-Fig.4 FTIR. Better to separate different spectra to clarify.

- Fig 5. We don´t know if they are talking about irradiation crosslining or temperature crosslinking or both, and put the units in axes X and Y.

-Fig 10. NMR, explain where we found R-N+H3 and put the tittle of the figure

-Page 12, line 261. Fig 12 shows that an increase in the NMA content in Thermal method

- Page 13 line 266 said: Irradiation lead to similar changes in the absorption spectra. But the authors didn´t explain before that they are talking about temperature crosslinking spectra.

-Fig 14 What does it means α in the figure? It is necessary to explain it immediately the equation or immediately in the text.

-page 14. Whit an increase of the hardener….What hardener??

-Conclusions, page 15, line 313 said thermal concentration. There are not thermal concentrations

Author Response

Dear Reviewer, we are very grateful for your Comments that allow us to improve the clearness and quality the Manuscript. We have done our best to fix everything in accordance with your recommendations.

The paper could be interesting, if the paper would be explain objectives, methodology and tittle of all figures better, it is very difficult to understand, the objective the experimental part and the conclusions, also the mechanisms; because they don’t explain if they are talking about UV or thermal reactions.

-Abstract. It said:  In this work the N-methylolacrylamide (NMA) and chitosan interaction under the action of UV radiation and heat treatment has been studied. But-In Introducction page 2 line 50-52 said: The purpose of this work is the development and scientific justification of a method for obtaining biocompatible chitosan hydrogels using NMA and modeling the curing of a polymer matrix under the action of ultraviolet radiation.  -It is not clear if they wanted compare heat with UV crosslinking reactions or they wanted to use both in the reaction.

The goal is correct. At the same time, the reactions occurring during the concentration of a polymer solution containing chitosan and NMA were studied, and it was shown that UV irradiation and subsequent concentration trigger the crosslinking process. The abstract has been corrected as following: “In this work, we studied the interaction of N-methylolacrylamide (NMA) and chitosan upon concentration of the solutions both under the action of UV radiation and without it, which results in curing of the polymer matrix.”

-Page 2 line 54 in section 2.2  It is not clear through the text if it was UV irradiated solutions of CS and also CS films. It is necessary to explain it well.

The prepared solutions only were exposed to UV radiation. This is indicated in section 2.5, and section 2.2 describes the process for preparing the initial solutions.

-Page 2, line 72. In which substrate?

Polystyrene Petri dish. Corrected.

-Page 3 it said. The films were made of all irradiated samples and the same samples but not UV irradiated. please clarify.

Thank you! We have replaced this sentence with the following: “The films made of them were compared with non-treated ones using relevant technical/spectroscopic data”.

-In other part of the text said modification of CS in solution is most effective carried out when polymer macromolecule forms a network. What kind of modification? Modification was carried out in solution and in film?, Explain better it through the text.

Presumably, the chemical interaction of NMA with chitosan begins from the moment of mixing these components in solution, intensifies as the solvent evaporates, and ends when the solvent is completely removed during film formation. In other words, an increase in the concentration of the polymer in solution is a factor that contributes to the effective interaction of NMA with chitosan. The polymer concentration needed to start the crosslinking can be determined from the viscosity concentration curves.

The beginning of the Results and Discussion unit has been rephrased as follows : “It is known that with an increase in the concentrations of reagents, the rate of their interaction increases, for bimolecular reactions exponentially. However, in photo-initiated reactions, the interaction scheme is complex, and the polymer concentration must be low for radiation to penetrate. The polymer concentration needed to start the crosslinking was determined from the viscosity concentration curves. The Crossover point (Figure 2) which separates section I of a weak dependence of viscosity on concentration from section II that is characterized by a smooth increase in viscosity in the region of 1.5-2.5 wt% due to structuring of solutions when the polymer macromolecules begin to associate.”

-page 3 line 92. All spectra were initially collected in ATR mode. Initially? and then in what mode?. And better to change the word collected by characterized or determine.

The obtained ATR spectra were converted into IR-Absorbance mode using a set of programs: Bruker Opus (version 6.1).

Page 4 line 130 said: The polymer concentration at which the macromolecules began to form an engagement network….Better to said: the polymer concentration needed to start the crosslinking.

-Page 4 line 131. The author said: increase the viscosity due to structuring of solutions’?’ it is necessary to clarify it.

-Fig 1 ( Moved to "Fig. S1") Author said. Dependence of viscosity on concentration from section II, characterized by smooth increase in viscosity due to structuring region??? The paragraph is not clear enough, it is necessary to explain better. What kind of structuring?. It is necessary to do the figure better.

Thank you. That was done as mentioned above.

-Page 5, line 148. NMA reacts with chitosan? How reacts? By crosslinking by copolymerization or what?

Both paths you mentioned. This is the subject of discussion below.

-Fig.4 ( Fixed to "Fig. 3") FTIR. Better to separate different spectra to clarify.

It seems to us that the chosen representation of the spectra gives more information about the change in the intensity of the bands. In significant regions, the spectra are separated quite distinctly.

- Fig 5. We don´t know if they are talking about irradiation crosslining or temperature crosslinking or both, and put the units in axes X and Y.

Thank you. All the discussed mechanisms relate to the interaction of polymers during the drying of the films (at concentrating of solutions at RT). We removed the wrong indication on heat treatment throughout the text and added the units in axes X and Y in the Figure.

Obviously, the mechanism at photoinitiation is different. It is difficult to explore when high molecular weight reagents are used. The films made of the irradiated solutions partially lost their solubility in dilute acid after holding for 48 h. Only the selected sol fraction (3-5 wt%) of the films was analyzed by NMR spectroscopy (Figure 9). The results as well as lost in solubility confirm the crosslinking reaction.

-Fig 10. (Fixed to "Fig. 9") NMR, explain where we found R-N+H3 and put the tittle of the figure

Thanks a lot. The Figure, text and the figure caption were corrected.

It was stated that «the films were washed with water to remove salts and with chloroform to remove NMA oligomers. The films partially lost their solubility (6-8% mass loss only was fixed) in dilute acid solutions, which confirms the crosslinking reaction. The sol fraction collected after washing the products with chloroform (3-5 wt%) was analyzed by NMR spectroscopy (Figure 9).”

The quality control of washing from salt (sodium acetate) was carried out using spectrophotometry by analyzing wash water. The sol fraction collected after washing the products with chloroform contains chitosan (in a basic form). These are non-crosslinked products, but they also indicate the presence of an interaction, since they contain chitosan, which is insoluble in chloroform if unmodified.

-Page 12, line 261. Fig 12 (Fixed to "Fig. 11") shows that an increase in the NMA content in Thermal method

Fig. 11. Absorption spectra of films from chitosan with content of NMA: 5 wt% (1, 1’); 10 wt% (2, 2’); 20 wt% (3, 3’); 30 wt% (4, 4’). 1-4 – before (without) UV exposure; 1’- 4’ – after UV exposure (1 h).

- Page 13 line 266 said: Irradiation lead to similar changes in the absorption spectra. But the authors didn´t explain before that they are talking about temperature crosslinking spectra.

We apologize for the misinformation about thermal crosslinking. All the previously considered mechanisms relate to the interaction of polymers during film drying (during concentration of solutions at RT). Corrected throughout.

-Fig 14 (Fixed to "Fig. 13") What does it means α in the figure? It is necessary to explain it immediately the equation or immediately in the text.

Done. Equation (2) with explanations is given in 2.10 section.

-page 14. Whit an increase of the hardener….What hardener??

Replaced with NMA.

-Conclusions, page 15, line 313 said thermal concentration. There are not thermal concentrations

Corrected. The conclusion has been carefully rewritten.

Reviewer 2 Report

The ms reports the interaction between N-methylolacrylamide (NMA) and chitosan under different conditions. The obtained photo-polymerized materials were characterized by a set of different techniques. My main concerns is that the purposed mechanism of interaction between Chi and NMA is made without any experimental support. Such a claim must be supported by experimental data related to intermediates. In my opinion, some of conclusions are not well supported by experimental data.

In the introduction authors should highlight the novelty of this work compared with those reported in references 13 and 14.

Authors have carried out UV-vis of Chi-NMA films; how did they do? An integrator sphere was used or the film was just placed in the cell at 45º? The procedure should be described to allow the reproducibility.

Is there anything new in Figure 2? The same with Figure 4 once no interaction occurs. In this case figure 4 can be displaced to SM without detriment.

Figure 13: it’s very hard to take any conclusion from this figure.

The magnitude of figures 14 must be reported.

How do authors have measured the swelling degree of chitosan with 0%NMD? The standard error associated to each data point should be plotted. Additionally, I can’t see any swelling kinetics data as indicated in the section 2.10.

Compare the behavior of films of crosslinked Chi with Chit (in powder state?) does not seems reliable.

Other minor points:

 “N-hydroxymethyl…”,”N-methylol”, etc. Following IUPAC rules,  “N” must be written in italic.

The supplier of acetic acid must be described

There are many figures some of them can be displaced to SM without detriment

Author Response

We are very grateful for your Comments that allow us to improve the clearness and quality the Manuscript. We have done our best to fix everything in accordance with your recommendations.

The ms reports the interaction between N-methylolacrylamide (NMA) and chitosan under different conditions. The obtained photo-polymerized materials were characterized by a set of different techniques. My main concerns is that the purposed mechanism of interaction between Chi and NMA is made without any experimental support. Such a claim must be supported by experimental data related to intermediates. In my opinion, some of conclusions are not well supported by experimental data.

We are aware that the proposed mechanisms are classical. It is difficult to prove them fully in the resulting cross-linked products (which, however, were carefully purified from unreacted non-cross-linked components), but we did not set such a goal. We offer possible reaction pathways based on known mechanisms, nothing more. This is not a discovery of mechanisms, but an explanation based on classical knowledge. We obtained the materials and carefully characterized their properties depending on the ratio of reagents (degree of swelling, morphology and mechanical characteristics). Both the IR and NMR spectra presented in the paper show that the supposed reactions have passed, and the products correspond to the hypothetical ones.

In the introduction authors should highlight the novelty of this work compared with those reported in references 13 and 14.

Thank you. The following phrase was added to Introduction before formulation of the purpose of the work : “However, a detailed description of possible ways of interaction upon concentrating of acidic aqueous solutions of chitosan containing NMA has not been considered previously.”

Authors have carried out UV-vis of Chi-NMA films; how did they do? An integrator sphere was used or the film was just placed in the cell at 45º? The procedure should be described to allow the reproducibility.

An integrating sphere was not used; a standard Shimadzu UV-2501PC dual-beam spectrophotometer was used; measurements were performed relative to air. 2.8 unit of the Materials and Methods was complemented accordingly.

Is there anything new in Figure 2? The same with Figure 4 once no interaction occurs. In this case figure 4 can be displaced to SM without detriment.

You are right. Both Figures were moved to SM.

Figure 13: it’s very hard to take any conclusion from this figure.

Of course, this is a qualitative analysis, which shows the changes in the intensity of the absorption bands, their shift and appearance a new absorption bands.  That may be associated with the appearance of new chromophores as a result of curing of the polymer matrix.

The magnitude of figures 14 must be reported.

Magnification was ×400. Added to 2.9 section. The micrographs show the size scale – 50 µm.

How do authors have measured the swelling degree of chitosan with 0%NMD? The standard error associated to each data point should be plotted. Additionally, I can’t see any swelling kinetics data as indicated in the section 2.10.

All measurements were done under similar experimental conditions as described in section 2.10. The title and the text were corrected since the data in the Figure present the maximum value of swelling degree only, you are right. The standard deviations (0.01 for 5 parallel measurements) were indicated in the Figure caption.

Compare the behavior of films of crosslinked Chi with Chit (in powder state?) does not seems reliable.

The film samples only were studied and compared.

Other minor points:

 “N-hydroxymethyl…”,”N-methylol”, etc. Following IUPAC rules,  “N” must be written in italic.

Done.

The supplier of acetic acid must be described

Done.

There are many figures some of them can be displaced to SM without detriment

Figures 2 and 4 were moved to SM.

Reviewer 3 Report

This is an interesting paper that contributes to the still-growing body of literature concerning chitosan and it's applications. However, there are a number of issues that the authors must address before the paper will be ready for publication:

1. In the Introduction the authors should specify precisely what applications of chitosan-based films require the formation of water-insoluble composites. For many medical applications including controlled drug release, a degree of solubility is very desirable. I realize that the authors are most likely focusing upon medical implants and associated devices but this needs to be clearly stated.

2. In Table 1 the Experiments should be labelled 'Experiment 1', 'Experiment 2', etc. not 'Experience' 1........2......3! In general, the paper needs a good proof read and correction by a fluent English speaker.

3. In equation 1 the * symbols must be replaced by . (implying multiplied by), or better still a proper 'multiplied by' symbol. 

4. In section 2.5 the photon flux of the lamp used must be stated. 

5. Figure 2 is not convincing. The graph does not show 2 clearly defined regions of differing slopes, it is a smooth curve. Given the likely variation in degree of deacetylation that exists with most chitosan starting materials supplied the result drawn from this graph is very questionable. The authors need to justify the use of this approach more clearly.

6. While the discussion of the possible reaction mechanisms is interesting, and most likely along the right lines, it is not really possible to identify either an SN1 or SN2 mechanism without more extensive detailed studies. Even if a proposed intermediate like the carbocation is identified (and it hasn't been) this is still not conclusive evidence of a mechanism. Any species seen under these conditions could simply be a 'spectator' species. Accordingly I would request that the discussion of the possible mechanisms be toned down a little. It is not really based on any data presented. 

7. Figure 13, the absorption spectra, are not very convincing given the likely variability of the chitosan starting material noted earlier. The variations noted are not really justified I feel. More discussion of the errors associated with sample-to-sample variation must be included alongside this figure. 

Author Response

We are very grateful for your Comments that allow us to improve the clearness and quality the Manuscript. We have done our best to fix everything in accordance with your recommendations.

This is an interesting paper that contributes to the still-growing body of literature concerning chitosan and it's applications. However, there are a number of issues that the authors must address before the paper will be ready for publication:

  1. In the Introduction the authors should specify precisely what applications of chitosan-based films require the formation of water-insoluble composites. For many medical applications including controlled drug release, a degree of solubility is very desirable. I realize that the authors are most likely focusing upon medical implants and associated devices but this needs to be clearly stated.

The Introduction was extended with following: “However, the material must be water insoluble (may be limited swelling), but degradable in a number of applications, namely tissue engineering constructions and surgical threads, in the case of textile threads, etc. To obtain water-insoluble chitosan products and control their plasticity, the polymer matrices should be modified.”

  1. In Table 1 the Experiments should be labelled 'Experiment 1', 'Experiment 2', etc. not 'Experience' 1........2......3! In general, the paper needs a good proof read and correction by a fluent English speaker.

Done. The paper was proof read and corrected thoroughly. 

  1. In equation 1 the * symbols must be replaced by (implying multiplied by), or better still a proper 'multiplied by' symbol.

Corrected.

  1. In section 2.5 the photon flux of the lamp used must be stated.

Done. The photon flux was 1×1017 photon / (cm2 × s).

  1. Figure 2 is not convincing. The graph does not show 2 clearly defined regions of differing slopes, it is a smooth curve. Given the likely variation in degree of deacetylation that exists with most chitosan starting materials supplied the result drawn from this graph is very questionable. The authors need to justify the use of this approach more clearly.

The paragraph was arranged as following: “The transition from the regime of dilute solutions to the moderately concentrated ones can be established more reliably using the modified Vinogradov–Pokrovskii method [15. Shipovskaya, A.B.; Abramov, A.Y.; Pyshnograi, G.V.; Al Joda, H.N.A. Rheological Properties of Aqueous Acid Solutions of Chitosan: Experiment and Calculations of the Viscometric Functions on the Basis of a Mesoscopic Model. J Eng Phys Thermophy 2016, 89, 642–651. https://doi.org/10.1007/s10891-016-1422-8]. However, to achieve the goals of this work, it is sufficient to establish an approximate range of concentrations for the formation of a fluctuation network. The polymer concentration needed to start the crosslinking when the polymer macromolecules begin to associate was determined from the viscosity concentration curves (Figure S1). It follows from the experiments that the concentration at which the network is formed corresponds to 2.0±0.2 wt%. Therefore, all further experiments were done with 2.0 wt% solutions of chitosan.”

The Figure was moved to Supplementary Materials.

  1. While the discussion of the possible reaction mechanisms is interesting, and most likely along the right lines, it is not really possible to identify either an SN1 or SN2 mechanism without more extensive detailed studies. Even if a proposed intermediate like the carbocation is identified (and it hasn't been) this is still not conclusive evidence of a mechanism. Any species seen under these conditions could simply be a 'spectator' species. Accordingly I would request that the discussion of the possible mechanisms be toned down a little. It is not really based on any data presented.

We are aware that the proposed mechanisms are classical. It is difficult to prove them fully in the resulting cross-linked products (which, however, were carefully purified from unreacted non-cross-linked components), but we did not set such a goal. We offer possible reaction pathways based on known mechanisms, nothing more. This is not a discovery of mechanisms, but an explanation based on classical knowledge. We obtained the materials and carefully characterized their properties depending on the ratio of reagents (degree of swelling, morphology and mechanical characteristics). The chemical structure of the resulting products was confirmed as far as it is possible using IR, UV and NMR spectroscopy. At the same time, understanding of the ongoing processes allows us to control the properties of the materials.

The conclusions have been expanded in view of the foregoing. Thanks for the suggestions.

  1. Figure 13, the absorption spectra, are not very convincing given the likely variability of the chitosan starting material noted earlier. The variations noted are not really justified I feel. More discussion of the errors associated with sample-to-sample variation must be included alongside this figure.

The presented absorption spectra make it possible to clearly see the trends, which are described in detail in the text after the figure: an increase in the NMA content is accompanied by a synchronous shift of the short-wavelength edge of the absorption band to the long -wavelength region (from 250 to 320 nm); after irradiation - a decrease in the intensity of the bands related to the original chitosan (in the region of 320–390 nm) and the appearance of new absorption bands reflecting ongoing crosslinking processes (an increase in the intensity of the bands in the region of 250–320 nm; a new absorption band appears in the region of 390–550 nm). The described trends in the change in the absorption spectra are observed for all the studied films. The Figure is used not for quantitative characteristics of the ongoing processes, but for qualitative observation of the absorption spectra of the films.

Reviewer 4 Report

This manuscript has serious scientific flaws. The assignment of most peaks in the spectra is wrong and therefore the interpretation of the spectra is wrong. The proposed reaction schemes are not supported by any data.

Due to the above I recommend rejection of this paper.

Author Response

This manuscript has serious scientific flaws. The assignment of most peaks in the spectra is wrong and therefore the interpretation of the spectra is wrong. The proposed reaction schemes are not supported by any data.

Due to the above I recommend rejection of this paper.

The IR spectra presented in the paper show that the supposed reactions have passed, and the products correspond to the hypothetical ones. The assignments of the absorption bands in the products are given in the work; the analysis was carried out on the basis of known literature data. NMR spectra were obtained for branched products without crosslinking. IR spectra make it possible to evaluate the structure of cross-linked products at different ratios of reagents.

We have extensive experience in working with the spectra of chitosan and its derivatives. This is a polymer with a complex structure (especially being cross-linked) and the characteristic bands may well be shifted. So, the band recorded by the spectrometer at 1560 cm-1 in the IR spectra (Fig. 5) belongs, in our opinion, to bending vibrations of primary amine groups, which appear at 1595-1600 cm-1 usually. This band definitely does not belong to the amide group, since the Amide II band cannot possibly be more intense than the one of Amide I. The extinction coefficients of the carbonyl (1650-60, C=O stretching vibrations, amide I) and NH bending, CN stretching (1550-60, amide II) bands differ greatly. The band at 1560 cm-1 has a shoulder at lower frequencies, which is Amide II, obviously. The intensity of this shoulder also increases with an increase in the amount of the reagent. δNH2 band appears at 1600 cm-1 as a shoulder to the intense absorption band of carbonyl stretching vibrations in the spectra of the products.

We are fully aware also that the proposed mechanisms are classical. It is difficult to prove them in the resulting cross-linked products (which, however, were carefully purified from unreacted non-cross-linked components), but we did not set such a goal. We offer possible reaction pathways based on known mechanisms, nothing more. This is not a discovery of mechanisms, but an explanation based on classical knowledge. At the same time, understanding of the ongoing processes allows us to control the properties of the materials. We obtained the materials and carefully characterized their properties depending on the ratio of reagents (degree of swelling, morphology and mechanical characteristics). To the best of our knowledge, these reactions have been considered in sufficient detail using the example of crosslinking of cellulose upon interaction with NMA. For chitosan, however, these reactions have not previously been considered in the literature in detail. We hope that such a detailed description of possible ways of interaction will be of interest to a wide range of readers.

We have done our best to correct the article in accordance with the recommendations of the reviewers. In view of the foregoing, we sincerely hope that you will change your mind about the possibility of publication.

Round 2

Reviewer 1 Report

The corrected manuscrit could be publish in the actually form.

Author Response

Thank you very much for your help and support.

Reviewer 2 Report

Authors have addressed my comments on a acceptable way.

Author Response

(The authors gave the same response as above.)

Reviewer 3 Report

Thank you for responding to my suggestions. In my view the paper is now suitable for publication

Author Response

(The authors gave the same response as above.)

Reviewer 4 Report

Unfortunately, I do not see any improvement of the manuscript.

The assignment of most peaks in the spectra is still wrong and therefore the interpretation of the spectra remains wrong.

In my opinion one cannot propose reaction schemes and mechanisms without sufficient data.

Due to the above I recommend rejection of this paper.

Author Response

Dear Reviewer,

We are unable to answer the remark about the assignment of spectral bands and their interpretation, since it is not formulated in detail. In our opinion, the spectral data are presented correctly. At your request and by agreement with the editor, we can move the schemes of the described classical mechanism of interaction to SI.